# Structural plasticity of dendritic secretory compartments during LTP-induced synaptogenesis

Yelena D Kulik[†], Deborah J Watson[‡], Guan Cao, Masaaki Kuwajima, Kristen M Harris*

Center for Learning and Memory, Department of Neuroscience, The University of Texas at Austin, Austin, United States

**Abstract** Long-term potentiation (LTP), an increase in synaptic efficacy following high-frequency stimulation, is widely considered a mechanism of learning. LTP involves local remodeling of dendritic spines and synapses. Smooth endoplasmic reticulum (SER) and endosomal compartments could provide local stores of membrane and proteins, bypassing the distant Golgi apparatus. To test this hypothesis, effects of LTP were compared to control stimulation in rat hippocampal area CA1 at postnatal day 15 (P15). By two hours, small spines lacking SER increased after LTP, whereas large spines did not change in frequency, size, or SER content. Total SER volume decreased after LTP consistent with transfer of membrane to the added spines. Shaft SER remained more abundant in spiny than aspiny dendritic regions, apparently supporting the added spines. Recycling endosomes were elevated specifically in small spines after LTP. These findings suggest local secretory trafficking contributes to LTP-induced synaptogenesis and primes the new spines for future plasticity.

DOI: https://doi.org/10.7554/eLife.46356.001

*For correspondence:
kmh2249@gmail.com

Present address: [†]Department of Biochemistry and Biophysics, Kavli Institute for Fundamental Neuroscience, University of California, San Francisco, San Francisco, United States; [‡]QPS, LLC Pencader Corporate Center, Newark, United States

Competing interests: The authors declare that no competing interests exist.

## Introduction

As the longest and most architecturally complex cells in the body, neurons face the unique challenge of regulating membrane and protein levels in distal compartments. Neurons have highly elaborate dendritic arbors. These dendrites possess synapses, points of contact where electrochemical transmission of information occurs. Most of the excitatory synapses are situated on dendritic spines, tiny protrusions with a head and neck comprising a geometry that is essential for shaping electrical signals (*Yuste and Denk, 1995*; *Hering and Sheng, 2001*; *Yuste, 2011*; *Harnett et al., 2012*; *Harris and Weinberg, 2012*; *Yuste, 2013*) and providing biochemical compartmentalization (*Harris and Stevens, 1989*; *Bourne and Harris, 2008*; *Colgan and Yasuda, 2014*). For synapses to function appropriately, the levels of receptor proteins at the postsynaptic density must also be finely tuned. Synapses are often located hundreds of micrometers away from the neuronal cell body. Adding to this spatial problem is the challenge of regulating protein abundance on the membrane in a temporally precise manner, as demanded by fast-acting processes such as synaptic potentiation.

Integral membrane proteins destined for the cell surface are canonically thought to be synthesized in the somatic rough endoplasmic reticulum, transported to the Golgi apparatus, and then secreted into the plasma membrane via exocytosis. It is now known that many proteins are translated locally in dendrites, a highly regulated process essential for normal development and plasticity (*Sutton and Schuman, 2006*; *Bramham and Wells, 2007*; *Hanus and Schuman, 2013*). Endoplasmic reticulum (ER) extends into dendrites, forming a continuous tubular network with regions of varying structural complexity and occasional entry into spines (*Spacek and Harris, 1997*; *Cooney et al., 2002*; *Cui-Wang et al., 2012*). Together with endosomes, the ER is perfectly positioned to provide

a local source of membrane and integral membrane proteins, such as glutamate receptors. However, the Golgi apparatus is absent in most distal dendrites. This puzzling observation has been resolved by recent work demonstrating that dendritic and somatic protein trafficking are highly segregated, and that glutamate receptors are trafficked through a specialized Golgi apparatus-independent pathway from the dendritic ER to the plasma membrane via recycling endosomes (*Bowen et al., 2017*). Structural changes in ER contribute to normal synaptogenesis during development and maturation (*Cui-Wang et al., 2012*). The involvement of this system in activity-induced synaptogenesis is unknown.

Long-term potentiation (LTP), the long-lasting enhancement of synaptic strength due to repetitive activity, is thought to underlie learning and memory. This process has been studied extensively in the hippocampus, a key brain region responsible for new memory formation. Insertion of glutamate receptors from an extrasynaptic reserve pool into the postsynaptic compartment is required for LTP in hippocampal area CA1 (*Granger et al., 2013*). LTP is also accompanied by structural changes in dendritic spines (*Bourne and Harris, 2012*; *Bailey et al., 2015*). In the young rat hippocampus, LTP produces new dendritic spines (*Watson et al., 2016*), contrasting with adult rat hippocampus where new spine outgrowth is stalled in favor of synapse enlargement (*Bourne and Harris, 2011*; *Bell et al., 2014*). While Golgi apparatus-independent trafficking has not been studied directly in the context of lasting LTP, recycling endosomes (RE) are known to supply AMPA receptors (*Park et al., 2004*), and recycling endosome exocytosis is required for spine formation and growth shortly after the induction of LTP (*Park et al., 2006*). Expanded knowledge about the involvement of Golgi apparatus-independent pathways in developmental synaptic plasticity could provide new targets for rescuing dysregulated synaptogenesis in cases of profound developmental disorders (*Fiala et al., 2002*).

Here, three-dimensional reconstruction from serial section electron microscopy (3DEM) revealed morphological changes in SER and endosomal compartments 2 hr following the induction of LTP. The findings are consistent with the involvement of the Golgi-bypass secretory system in supporting synaptic plasticity in the developing hippocampus.

## Results

An acute within-slice experimental protocol (*Watson et al., 2016*) was used to compare the effects of TBS and control stimulation on subcellular membranous compartments in dendrites. In brief, two stimulating electrodes were positioned ~800 µm apart with a recording electrode halfway in between them in CA1 stratum radiatum of P15 rat hippocampus in one slice from each of two animals (*Figure 1A*). Baseline responses were collected from both electrodes. TBS was delivered at one stimulating electrode and control stimulation was delivered at the other stimulating electrode, counterbalanced in position relative to CA3 for each experiment. There was a significant increase in the field excitatory postsynaptic potential (fEPSP) slope immediately after TBS (*Figure 1B,C*). Slices were fixed 120 min later. EM image volumes were collected from tissue on a diagonal ~120 µm below and to the side of each stimulating electrode. Segments of spiny dendrites, synapses, and all subcellular membrane compartments were reconstructed in three dimensions (see Materials and methods for details).

### Limited entry of SER into dendritic spines

Consistent with previous reports on hippocampal dendrites (*Spacek and Harris, 1997*; *Cooney et al., 2002*), the SER formed an anastomosing network throughout the dendritic shaft with occasional entry into a subset of dendritic spines (*Figure 2A*; see *Figure 2—figure supplement 1* for all analyzed dendrites reconstructed with SER). While the dendritic spine density more than doubled 2 hr following TBS, a similar increase in the occurrence of SER in spines did not occur (*Figure 2B,C*).

Spines with small synapses, as measured by the surface area of the postsynaptic density (PSD) (<0.05 µm$^2$), accounted for the LTP-induced increase in spine density (*Figure 2B*). This difference was not present at earlier times, and the small spines more than tripled in density by 2 hr post induction of LTP, suggesting that most of this population comprised newly formed spines (*Watson et al., 2016*). There were no significant effects on SER content in these small spines; not in frequency of spine-localized SER (*Figure 2D*), average SER volume (*Figure 2E*), nor average SER surface area

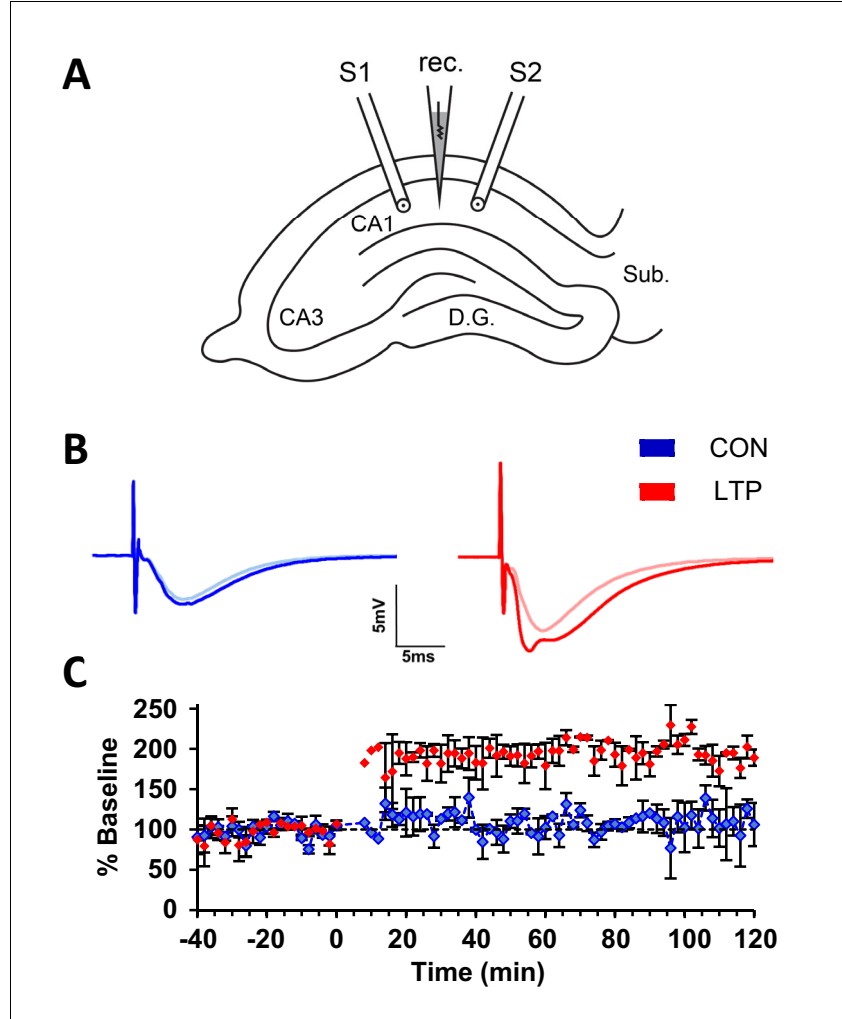

**Figure 1.** Within-slice experimental design and electrophysiological outcome. (**A**) Illustration of an acute slice from a P15 rat hippocampus with a recording electrode (rec.) in the middle of CA1 stratum radiatum between two bipolar stimulating electrodes (S1 and S2). S1 and S2 are separated by 600-800 μm. The two experiments were counterbalanced for which of the two electrodes delivered TBS or control stimulation. Tissue samples collected for 3DEM were located ~120 μm beneath and to the side of the stimulating electrodes. D.G., dentate gyrus; Sub., subiculum. (**B**) Representative waveforms from control (CON, blue) and TBS (LTP, red) sites. Each waveform is the average of the final 10 responses to each stimulating electrode obtained for the last 20 min before delivery of TBS at *time 0* (light color) and for 20 minutes before the end of the experiment at 120 min after TBS (dark color). The stimulus intensity was set at population spike threshold to activate a large fraction of the axons in the field of each stimulating electrode. The positive deflection in the post-TBS waveform at ~3-4 ms reflects synchronous firing of pyramidal cells with LTP. (**C**) Changes in the slope of the field excitatory postsynaptic potential (fEPSP), expressed as a percentage of the average baseline response to test-pulses, were recorded for 20 min before delivery of TBS at *time 0* (red) or control stimulation (blue). Responses were recorded for n=2 slices for 120 min after the first TBS train, then fixed and processed for 3DEM as described in Methods. Error bars are SEM. Adapted from *Watson et al. (2016)* where it was originally published under a CC BY-NC-ND 4.0 license https://creativecommons.org/licenses/by-nc-nd/4.0/).

DOI: https://doi.org/10.7554/eLife.46356.002

The following source data is available for figure 1:

**Source data 1.** Excel spreadsheet containing the raw numbers that generated the graphs and waveforms for these experiments.

DOI: https://doi.org/10.7554/eLife.46356.003

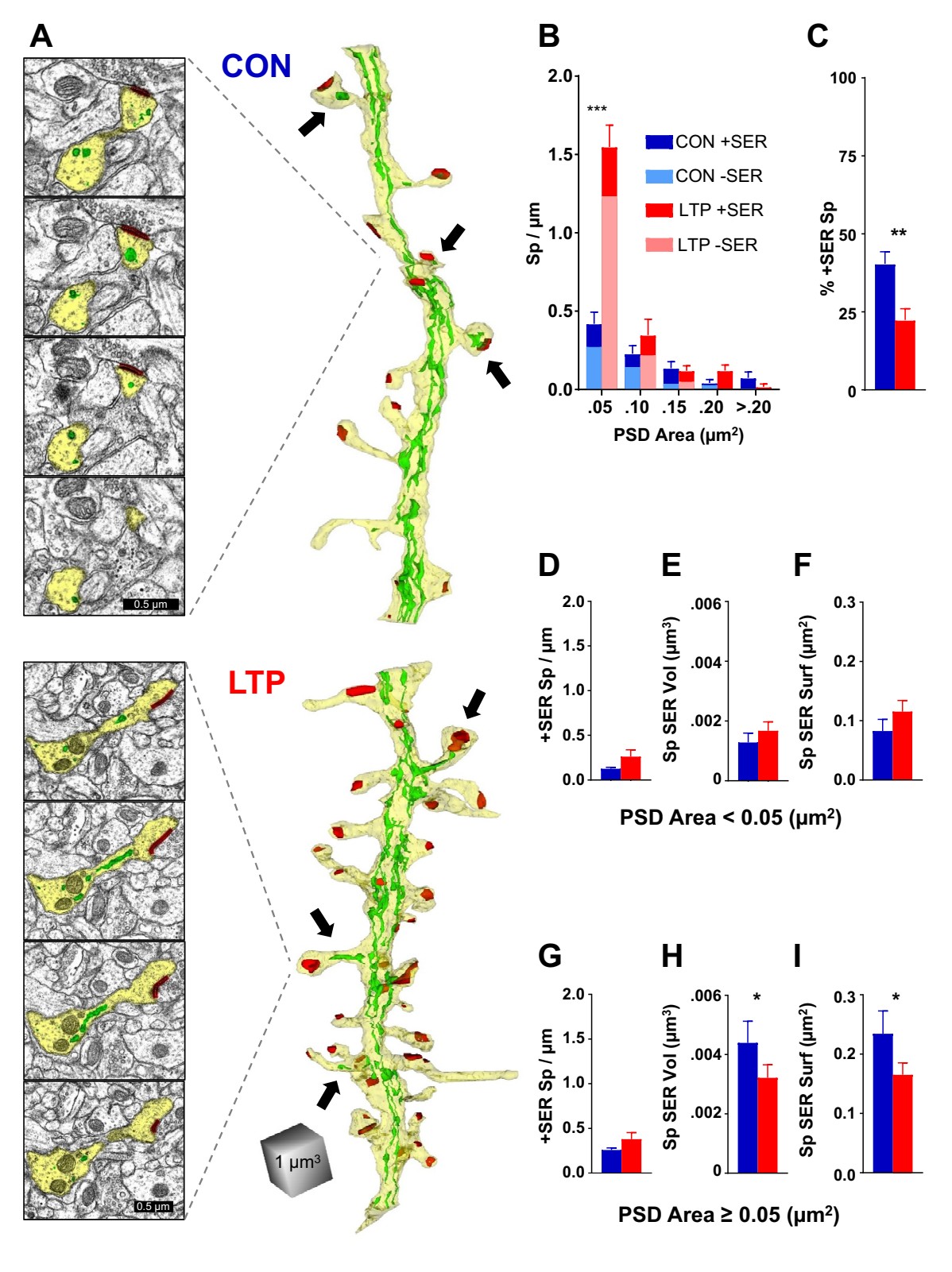

**Figure 2.** The limited occupancy of spines by SER does not increase during spinogenesis in the LTP condition. (A) Sample serial section EMs (left) and representative 3D reconstructions of dendrites (right) from control (top) and LTP (bottom) conditions, illustrating dendrites (yellow), SER (green), and synapses (red). Synaptic area was measured as the total surface area of the PSD. Arrows point to SER-containing spines. (B) Spine density (#/μm) binned for PSD area. Significant increase in spines following TBS was carried by spines in the category with the smallest PSD areas (<0.05 μm$^2$; ANOVA $F_{(1,12)}$
*Figure 2 continued on next page*

*Figure 2 continued*

=50.707, p=0.00001, $\eta^2$ = 0.81). No statistically significant changes occurred in the frequency of spines with larger synapses (PSD area 0.05 to 0.1 $\mu m^2$, ANOVA $F_{(1,12)}$=1.079, p=0.31941; PSD area 0.1 to 0.15 $\mu m^2$, ANOVA $F_{(1,12)}$=0.09638, p=0.76154; PSD area 0.15 to 0.2 $\mu m^2$, ANOVA $F_{(1,12)}$=3.5065, p=0.08569; PSD area >0.2 $\mu m^2$, ANOVA $F_{(1,11)}$=3.0778, p=0.10484). Control n = 8, LTP n = 8 dendrites. (C) Decrease in percentage of spines containing SER following TBS (ANOVA $F_{(1,12)}$=10.599, p=0.00688, $\eta^2$ = 0.87). Control n = 8, LTP n = 8 dendrites. (D–F) SER content for spines with PSD areas less than 0.05 $\mu m^2$. (D) No statistically significant difference between control and LTP conditions in density of spines with SER (ANOVA $F_{(1,12)}$=2.59, p=0.13322). Control n = 8, LTP n = 8 dendrites. (E) No statistically significant difference in average SER volume per SER-containing spine between control and LTP conditions (hnANOVA $F_{(1,14)}$=.73111, p=0.40692). Control n = 12, LTP n = 15 spines. (F) No statistically significant difference in SER surface area per SER-containing spine between control and LTP conditions (hnANOVA $F_{(1,14)}$=3.3120, p=0.09022). Control n = 12, LTP n = 15 spines. (G–I) SER content for spines with total PSD area equal to or greater than 0.05 $\mu m^2$. (G) No statistically significant difference in density of spines with SER between control and LTP conditions (ANOVA $F_{(1,12)}$=2.1641, p=0.16700). Control n = 8, LTP n = 8 dendrites. (H) Reduction in average SER volume per SER-containing spine in the LTP relative to control condition (hnANOVA $F_{(1,38)}$=5.7205, p=0.02182, $\eta^2$ = 0.13). Control n = 29, LTP n = 25 spines. (I) Reduction in average SER surface area in SER-containing spines in the LTP relative to control condition (hnANOVA $F_{(1, 38)}$=4.5873 p=0.03868, $\eta^2$ = 0.12). Control n = 29, LTP n = 25 spines. Bar graphs show mean ± S.E.M. Control (CON, blue) and TBS (LTP, red).

DOI: https://doi.org/10.7554/eLife.46356.004

The following source data and figure supplement are available for figure 2:

**Source data 1.** Excel spreadsheets containing the raw numbers that generated the graphs in each part of this figure along with the summary of statistics.
DOI: https://doi.org/10.7554/eLife.46356.006
**Figure supplement 1.** All analyzed dendrites fully reconstructed with SER, aligned left to right from least to greatest spine density.
DOI: https://doi.org/10.7554/eLife.46356.005

(*Figure 2F*). Since the occurrence of SER did not keep pace with the increase in small spines, the most parsimonious interpretation is that the LTP-induced new spines did not acquire SER.

In contrast, while the incidence of SER entry into spines with larger synapses (PSD area $\geq 0.05$ $\mu m^2$) did not change (*Figure 2G*), there was however a decrease in the average volume (*Figure 2H*) and surface area (*Figure 2I*) of SER in these spines. The spine apparatus is an organelle comprising cisterns of SER laminated with electron dense plates that may serve Golgi functions in spines (*Gray, 1959*; *Špaček, 1985*; *Pierce et al., 2001*). Consistent with previous observations (*Spacek and Harris, 1997*; *Cooney et al., 2002*), the spine apparatus appeared in only one dendrite in each condition (data not shown), suggesting that this structure is not central to the activity-induced spinogenesis at this age. Overall, these results reveal that SER entry into dendritic spines is limited and does not scale up with rapid synaptogenesis following LTP at P15.

## Reduced complexity in shaft SER after LTP

Previous work demonstrated in cultured neurons that local zones of ER complexity produce ER exit sites and compartmentalize membrane proteins near the base of dendritic spines (*Cui-Wang et al., 2012*). Consistent with this finding, SER was inhomogeneously distributed across spiny and aspiny regions of the dendrites in both control and LTP conditions. SER appeared as small circular profiles on some sections, and swollen cisternae with bridging elements on other sections (*Figure 3A*). In 3D reconstruction, the primarily tubular structure of SER in aspiny regions and the expanded SER in spiny regions of the dendrite become apparent (*Figure 3B*). Following LTP, there was a trend towards reduced shaft SER surface area (*Figure 3C*) that reached statistical significance with reduced shaft SER volume (*Figure 3D*) when quantified across the total length of the dendritic segments. The SER complexity was estimated by summing the dendritic shaft SER cross-sectional areas in each section, assigning the value to the spiny or aspiny segments, and summing across their independent lengths (*Cui-Wang et al., 2012*). This measure of SER complexity was greater in spiny than aspiny segments under both conditions yet was significantly reduced in both the aspiny and spiny regions following LTP relative to the control condition (*Figure 3E*). Considering the prior work, this outcome suggests that SER resources may have contributed to the spine outgrowth by 2 hr following the induction of LTP.

## Identifying the dendritic trafficking network

Recent work has shown that SER participates in a local, Golgi apparatus-independent secretory trafficking pathway through recycling endosomes in dendrites (*Bowen et al., 2017*). Recycling endosomes have been identified as transferrin receptor-positive membrane compartments in dendrites

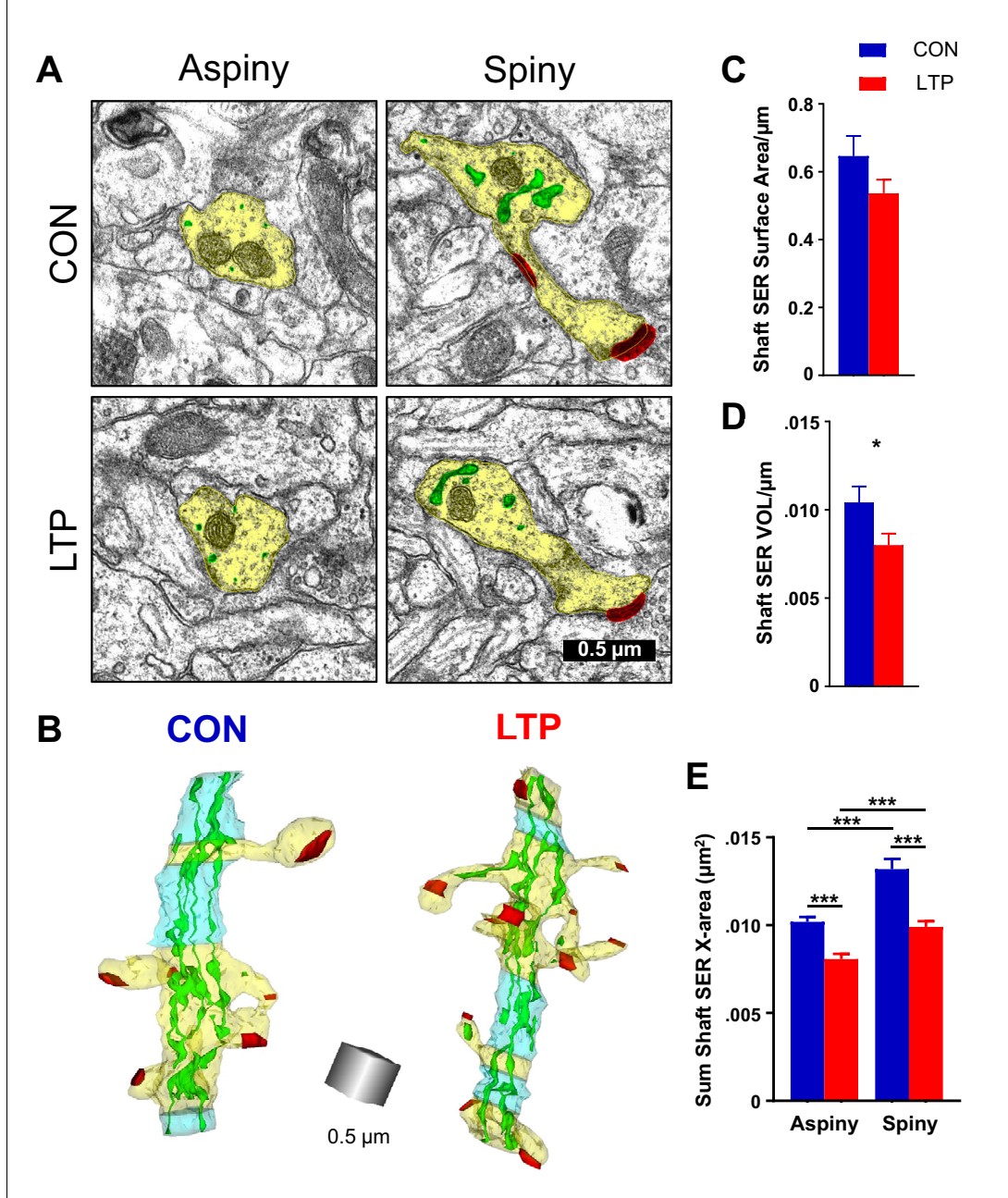

**Figure 3.** Reduction in shaft SER following LTP. (A) Electron micrographs showing the dendrite (yellow), SER (green), and synapses (red). For both control and LTP, the SER in the aspiny segments forms small cross-sectioned tubules, whereas in the spiny segments the SER tubules are broadly expanded. (B) Sample 3D reconstructions from serial section electron micrographs of SER-containing dendrites, illustrating spiny segments (yellow) and aspiny segments (blue) while the other colors match *Figure 2*. Aspiny segments consist of two or more sections (>100 nm) of no spine origins. Spiny segments had at least one spine and were surrounded by aspiny segments. Scale cube is 0.5 μm on each side. (C) No statistically significant differences between control and LTP conditions were found in surface area of SER in the dendritic shaft (ANOVA $F_{(1,12)}$=3.8833, p=0.07228). Control n = 8, LTP n = 8 dendrites. (D) Volume of dendritic SER network was reduced in the LTP relative to control conditions (ANOVA $F_{(1,12)}$=6.4397, p=0.02605, $\eta^2$ = 0.35). Control n = 8, LTP n = 8 dendrites. (E) Summed cross-sectional area of SER tubules and cisterns as a measure of changes in complexity. More SER on spiny than aspiny sections within both control (hnANOVA $F_{(1,1432)}$ = 51.672, p<0.00000, $\eta^2$ = 0.034; spiny n = 493, aspiny n = 955 sections) and LTP conditions (hnANOVA $F_{(1,324)}$=17.535, p=0.00003, $\eta^2$ = 0.013; spiny n = 714, aspiny n = 626 sections). Reduced SER complexity with LTP for both spiny (hnANOVA $F_{(1,1191)}$ = 51.745, p<0.00000, $\eta^2$ = 0.019; Control n = 493, LTP n = 714 sections) and aspiny sections (hnANOVA $F_{(1,1565)}$ = 29.991, p<0.00000, $\eta^2$ = 0.042; Control n = 955, LTP n = 626 sections) relative to control. Bar graphs show mean ± S.E.M. Control (CON, blue) and TBS (LTP, red).

DOI: https://doi.org/10.7554/eLife.46356.007

*Figure 3 continued*

The following source data is available for figure 3:

**Source data 1.** Excel spreadsheets containing the raw numbers that generated the graphs in each part of this figure along with the summary of statistics.
DOI: https://doi.org/10.7554/eLife.46356.008

by immuno-EM (*Park et al., 2006*). Other work found that non-SER subcellular components endocytose BSA-conjugated gold particles from the extracellular space (*Cooney et al., 2002*). Together these findings suggest that while these two compartments interact, the SER is not an endocytic structure. Here we considered the possibility that the endosome-based satellite system was also mobilized during LTP.

Once the continuous network of SER was reconstructed, the non-SER compartments could be identified as distinct terminating structures. Endosomal subtypes were classified as depicted in *Figure 4A* (*Cooney et al., 2002*; *Park et al., 2006*; *Deretic, 2008*; *Von Bartheld and Altick, 2011*). Coated pits, coated vesicles, and large vesicles were treated as one category of primary endocytic structures. Sorting complexes and recycling complexes were treated as functionally separate categories. Whorls, free multivesicular bodies, lysosomes, and autophagosomes were classified as degradative structures. Detailed descriptions based on EM morphology follow.

Tubules were cylindrical in shape with a smooth outer membrane, uniform diameter, and a dark, grainy interior. When two or more tubules occurred in proximity, they were categorized as a recycling complex (*Figure 4B*; *Figure 4—figure supplement 1*, *Figure 4—video 1*). Vesicles were distinguished from tubules by examining adjacent sections. Small vesicles (40–60 nm diameter, *Figure 4B*; *Figure 4—figure supplement 1*) and large vesicles (60–95 nm diameter) had a smooth outer membrane and ended within 1–2 sections. Coated pits were omega-shaped invaginations surrounded by clathrin coats (*Figure 4C*; *Figure 4—figure supplement 2*). Coated vesicles had a clathrin coat, were free-floating in the cytoplasm. Occasionally, clathrin-coated buds were observed on the ends of tubules.

Multivesicular bodies (MVB) contained a variable number of internal vesicles. When a multivesicular body was found surrounded by tubules, the grouping was categorized as a sorting complex (*Figure 4D*; *Figure 4—figure supplement 3* and *Figure 4—video 2*). Future work might reveal some MVBs to be exosomal compartments (*Ashley et al., 2018*; *Pastuzyn et al., 2018*). Amorphous vesicles had a smooth membrane, an electron-lucent interior, and an irregular shape (*Figure 4E*; *Figure 4—figure supplement 4*).

Lysosomes were spherical structures with a homogeneous, electron-dense interior enclosed by one membrane and measuring 70–150 nm in diameter (*Figure 4F*; *Figure 4—figure supplement 5*). Lysosomes were classified as degradative structures. A MVB was considered to be a primary lysosome, namely a degradative structure, when found alone and containing vesicles or pieces of membrane in a dark matrix (*Parton et al., 1992*; *Futter et al., 1996*; *Mukherjee et al., 1997*; *Cooney et al., 2002*). Whorls had multiple convoluted membranes spanning many sections, had a single point of entry into the dendrite, and were classified as degradative structures (*Figure 4G*; *Figure 4—figure supplement 6*; *Figure 4—video 3*). All non-degradative structures were classified as constructive for the quantitative analyses presented next.

## Constructive endosomes occurred more frequently in spines after LTP

Endosomal structures occurred in the dendritic shafts and a subset of spines (*Figure 5A*; see *Figure 5—figure supplement 1* for all analyzed dendrites reconstructed with constructive endosomes). Overall, endosomal frequency did not change significantly across conditions within dendritic shafts (*Figure 5B*); however, when analyzed by subtype the occurrence of recycling complexes was increased (*Figure 5B*). Similarly, there was no significant effect of LTP relative to the control condition on endosomal distribution to aspiny or spiny dendritic segments.

In contrast, there was a substantial increase in the occurrence of dendritic spines with endosomes, an effect that was confined to spines with small PSD areas (<0.05 $\mu m^2$, *Figure 5A,C,D*). Furthermore, this increase in spines involved constructive endocytic compartments (including coated pits, coated vesicles, large vesicles, recycling complexes, and small vesicles), with no significant effects on the rare occurrence of spines with amorphous vesicles, sorting complexes, or degradative structures

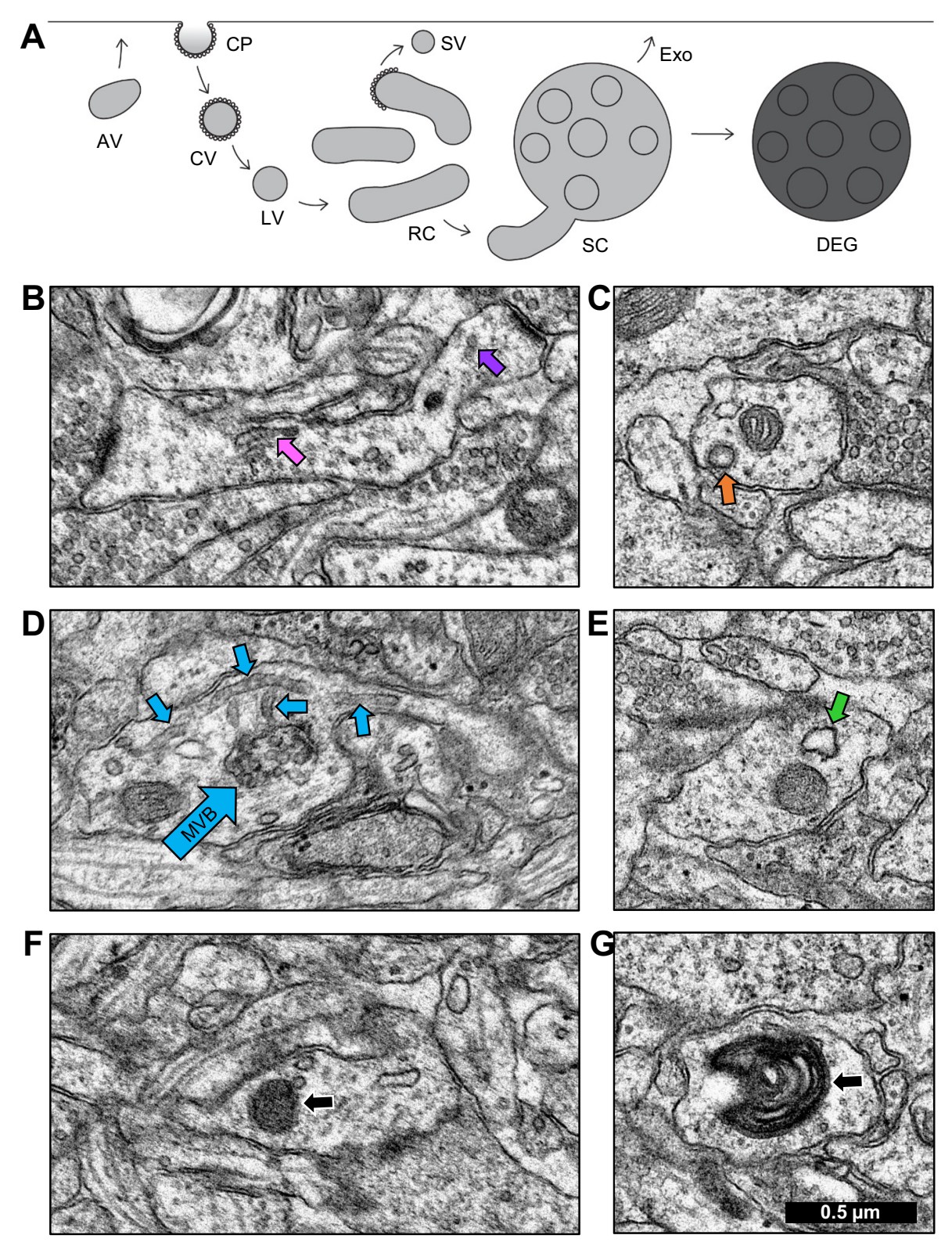

**Figure 4.** Identification of endosomal compartments. (A) Model of the dendritic endosomal pathway. Clathrin-coated pits (CPs) invaginate, becoming clathrin-coated vesicles (CVs) and large vesicles (LVs) after coat shedding. Large vesicles fuse to form tubules, recycling complexes (RCs), and sorting complexes (SCs) with a multivesicular body (MVB). From here, the sorted material may be sent to the plasma membrane via small vesicles (SVs) that pinch off coated tips of tubules. MVBs may serve as exosomes (Exo) or primary lysosomes, that are more darkly stained than exosomes due to the

*Figure 4 continued on next page*

*Figure 4 continued*

acidic cytomatrix of lysosomes (adapted from *Cooney et al., 2002*). Sample electron micrographs illustrate (**B**) recycling complex (pink arrow) and small vesicles (purple arrow), (**C**) clathrin-coated pit (orange arrow), (**D**) sorting complex (light blue arrows point to multivesicular body (MVB) in the center and tubules around it), (**E**) amorphous vesicle (green arrow), (**F**) lysosome (black arrow), and (**G**) whorl (black arrow). Scale bar in (**G**) is 0.5 μm for all images.
DOI: https://doi.org/10.7554/eLife.46356.009

The following video, source data, and figure supplements are available for figure 4:

**Source data 1.** Excel spreadsheets containing details of the locations of each object in *Figure 4*.
DOI: https://doi.org/10.7554/eLife.46356.016

**Figure supplement 1.** Sample images from the LTP condition of dendritic (yellow) recycling complex with multiple tubules (pink) entering the spine neck, and two small vesicles (purple arrow) in a different dendritic spine.
DOI: https://doi.org/10.7554/eLife.46356.010

**Figure supplement 2.** Sample images of coated pit (orange) inside the dendritic shaft (yellow) from a dendrite in the LTP condition, D25 DCPBM sections 121–124.
DOI: https://doi.org/10.7554/eLife.46356.011

**Figure supplement 3.** Sample images of sorting complex (turquoise) inside the dendritic shaft from the control condition, with one tubule entering a spine neck (right side row 3).
DOI: https://doi.org/10.7554/eLife.46356.012

**Figure supplement 4.** Sample image of an amorphous vesicle in the dendritic shaft of the LTP condition from D35 DCPBM sections 25–28.
DOI: https://doi.org/10.7554/eLife.46356.013

**Figure supplement 5.** Sample images of degradative lysosome (black) in the dendritic shaft (yellow) of the LTP condition from D17 FZYJV sections 146–149.
DOI: https://doi.org/10.7554/eLife.46356.014

**Figure supplement 6.** Sample images of degradative whorl (black) in a dendrite (yellow) of the control condition from D69 FXBVK sections 176–180.
DOI: https://doi.org/10.7554/eLife.46356.015

**Figure 4—video 1.** Video paging through dendritic from the LTP condition including sections 96–121 of D28 FZYJV.
DOI: https://doi.org/10.7554/eLife.46356.017

**Figure 4—video 2.** A sorting complex (turquoise) in a dendrite (yellow) from the control condition is D26 PWCNZ and includes sections 35–46.
DOI: https://doi.org/10.7554/eLife.46356.018

**Figure 4—video 3.** A degradative whorl (black) in a dendrite of the control condition from D69 FXBVK sections 170–187.
DOI: https://doi.org/10.7554/eLife.46356.019

(*Figure 5E*; see *Figure 5—figure supplement 2* for all analyzed dendrites reconstructed with degradative endosomes). These data suggest that the non-canonical secretory trafficking contributes locally in support of spines added 2 hr following the induction of LTP at P15.

## Discussion

These results provide several advances towards understanding mechanisms of enduring LTP in the developing hippocampus. A population of spines that increased in density by 2 hr after the induction of LTP relative to control stimulation had small synapses and mostly lacked SER. Spines with larger synapses were unchanged in density and retained SER in similar proportions under both conditions. The distribution of SER along the dendritic shaft was non-uniform, with greater abundance and complexity in spiny than aspiny regions under control and LTP conditions. However, the shaft SER was reduced in volume and complexity after LTP. In conjunction, there was an LTP-related increase in endosomal structures confined to the small, presumably newly formed spines. This elevation involved constructive endocytic, recycling, and exocytic structures in the small spines. In contrast, no differences occurred between control and LTP conditions in the frequency or locations of the degradative structures.

These data are from two animals using the within-slice paradigm to control for between-slice variance. The stimulating electrodes were positioned such that the sampling of dendrites was counterbalanced with respect to position from the CA3 axons that were stimulated. Dendrites were matched for caliber to avoid the confound that thicker dendrites have more spines per micron. Future work will be needed to determine whether these findings generalize beyond the medium caliber dendrites and position within the dendritic arbor, and to other slice and LTP induction paradigms.

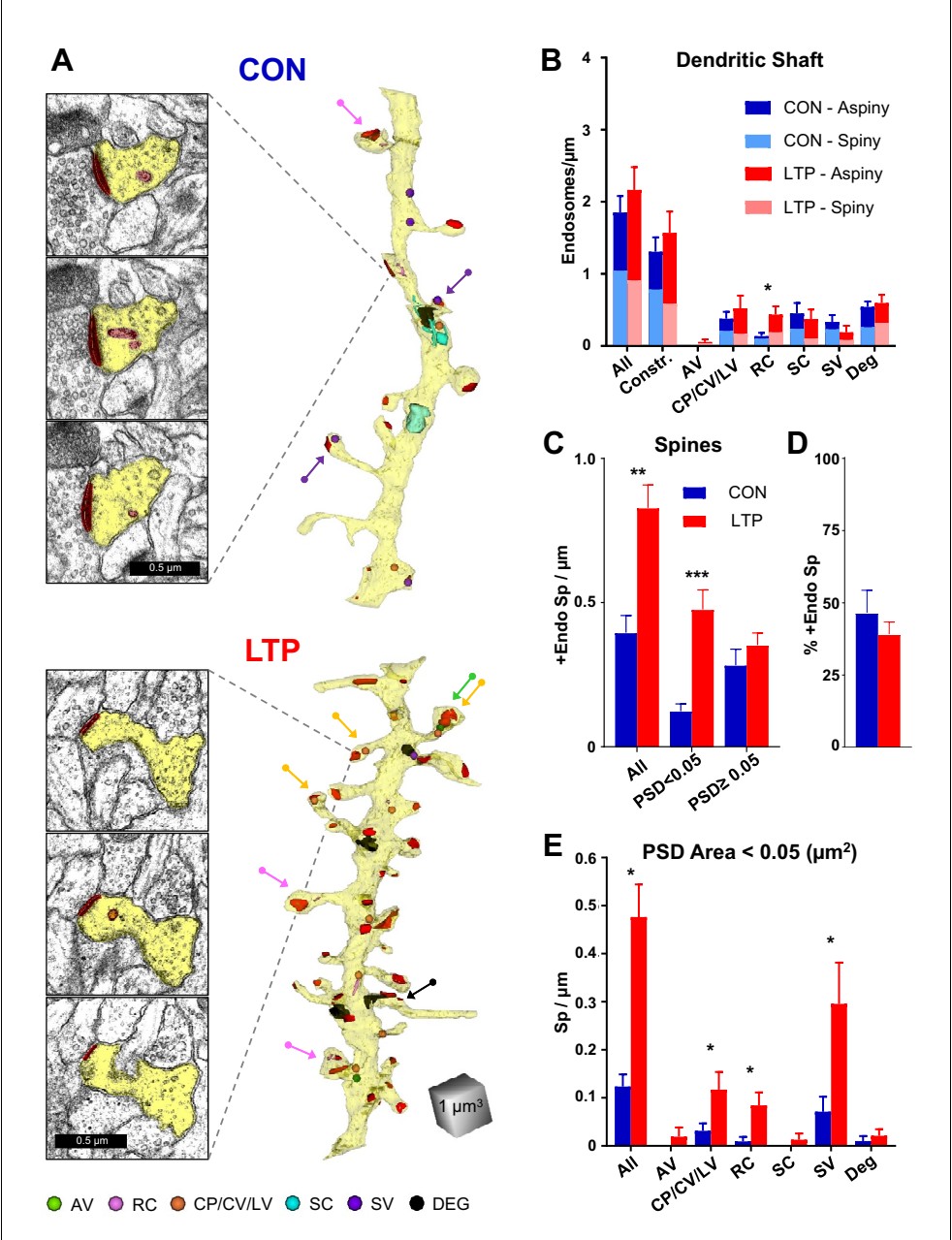

**Figure 5.** Increased occurrence of endosomes in small spines after LTP. (**A**) Sample serial EM sections and representative 3D reconstructed dendrites illustrate the distribution of endosomal compartments from control and LTP conditions. Dendrites are yellow, synapses are red, and color-coded arrows point to endosome-containing spines. The color-coded key in the lower left corner indicates amorphous vesicles (AV), recycling complexes (RC), coated pits (CP), coated vesicles (CV), large vesicles (LV), sorting complexes (SC), small vesicles (SV) and degradative structures (DEG); these abbreviations apply also to the graphs. Vesicles are represented as 100 nm spheres (AV, CP, CV, LV, and SV). The other structures (RC, SC, DEG) are reconstructed in 3D to scale. (**B**) Endosomal structures in dendritic shafts (#/μm) with relative distributions to aspiny and spiny segments in control (CON) and LTP conditions. Overall, shaft endosomes (hnANOVA $F_{(1,293)}$=0.93104, p=0.33539), degradative structures (hnANOVA $F_{(1,293)}$=0.47789, p=0.48993) or constructive endosomal compartments (Constr. = all minus degradative; hnANOVA $F_{(1,293)}$=0.62167, p=0.43107) did not differ between LTP and control conditions or segment locations. Recycling complexes (RC) were greater in the LTP than control dendritic shafts (hnANOVA $F_{(1,293)}$=6.4920, p=0.01135, $\eta^2$ = 0.022), but no significant differences occurred in the other categories: amorphous vesicles (hnANOVA $F_{(1,293)}$=1.5092, p=0.22025); small vesicles (hnANOVA $F_{(1,293)}$=1.1699, p=28031); coated pits, coated vesicles, and large vesicles (hnANOVA $F_{(1,293)}$=0.89152, p=0.34584); and sorting complexes (hnANOVA $F_{(1,293)}$=0.45286, p=0.50151). (For control (CON) n = 151 aspiny + spiny segments and for LTP n = 158 aspiny + spiny segments.) (**C**) More dendritic spines contained endosomes along the dendrites in the LTP than the control condition (ANOVA $F_{(1,12)}$=18.047, p=0.00113, $\eta^2$ = 0.60), an effect that was carried by spines with PSD areas less than 0.05 μm$^2$ (ANOVA $F_{(1,12)}$=23.642, p=0.00039, $\eta^2$ = 0.66) but not in spines with PSD area $\geq$0.05 μm$^2$ (ANOVA $F_{(1,12)}$=0.84714, p=0.37550). (**D**) Stability in percentage of spines containing endosomes following TBS (ANOVA $F_{(1,12)}$=.72158, p=0.41225).

*Figure 5 continued on next page*

*Figure 5 continued*

(**E**) Among spines with PSD area less than 0.05 μm$^2$, the increase in occupancy of endosomes was due to more with coated pits, coated vesicles, and large vesicles (ANOVA $F_{(1,12)}$=4.94433, p=0.046140, $\eta^2$ = 0.29), recycling complexes (ANOVA $F_{(1,12)}$=11.009, p=0.00613, $\eta^2$ = 0.48), and more with small vesicles (ANOVA $F_{(1,12)}$=5.2575, p=0.04072, $\eta^2$ = 0.30). No significant changes in spine occupancy occurred for amorphous vesicles (ANOVA $F_{(1,12)}$=1, p=0.33705), sorting complexes (ANOVA $F_{(1,12)}$=1, p=0.33705), or degradative structures (ANOVA $F_{(1,12)}$=0.46689, p=0.5074). Bar graphs show mean ± S.E.M. (For **C–E**), Control (CON, n = 8 full dendrite reconstructions) and LTP (n = 8 full dendrite reconstructions).

DOI: https://doi.org/10.7554/eLife.46356.020

The following source data and figure supplements are available for figure 5:

**Source data 1.** Excel spreadsheets containing the raw numbers that generated the graphs in each part of this figure along with the summary of statistics.
DOI: https://doi.org/10.7554/eLife.46356.023
**Figure supplement 1.** All analyzed dendrites fully reconstructed with constructive endosomes, aligned left to right from least to greatest spine density.
DOI: https://doi.org/10.7554/eLife.46356.021
**Figure supplement 2.** All analyzed dendrites fully reconstructed with intracellular degradative structures, aligned left to right from least to greatest spine density.
DOI: https://doi.org/10.7554/eLife.46356.022

The findings suggest a model in which local Golgi apparatus-independent secretory trafficking adds and prepares new spines for subsequent plasticity (*Figure 6*). TBS induces LTP via the insertion of glutamate receptors from recycling endosomes and lateral diffusion (*Malinow and Malenka, 2002*; *Choquet and Triller, 2013*). By 5 min (early LTP), there is a temporary swelling of spines and recycling endosomes are recruited into the spines; however the PSD is not enlarged at this early timepoint suggesting receptors are inserted into pre-existing slots (*Park et al., 2004*; *Lisman and Raghavachari, 2006*; *Park et al., 2006*; *Bourne and Harris, 2011*; *MacGillavry et al., 2013*; *Watson et al., 2016*). By two hours (late LTP), shaft SER decreases as it contributes membrane and proteins via ER exit sites to the formation of new spines, which have silent synapses lacking AMPAR. Constructive endosomes are recruited to the new spines and provide a reserve pool of receptors that are in position for rapid insertion of AMPAR upon subsequent potentiation.

## Effects of LTP on SER and spines

Previous work has shown that integral membrane proteins rapidly diffuse throughout tubular SER and become confined in regions where the SER is more complex, having branches between tubules and distended cisternae (*Cui-Wang et al., 2012*). As spine density increases across development so too does SER complexity, leading to decreased mobility of ER membrane cargo with age. SER complexity was measured as the summed cross-sectional area to capture the local variation. SER and spine density were positively correlated where more dendritic spines clustered locally. Using the same methods, we found SER volume and complexity were greater in spiny than aspiny regions and were reduced in conjunction with TBS-induced spinogenesis along these P15 dendrites. This result suggests that the membrane lost from SER in the shaft could have been used to build new spines after LTP.

In adult hippocampal area CA1, LTP produced synapse enlargement at the expense of new spine outgrowth (*Bourne and Harris, 2011*; *Bell et al., 2014*; *Chirillo et al., 2019*). SER is a limited resource, entering only 10–20% of hippocampal dendritic spines (*Spacek and Harris, 1997*; *Cooney et al., 2002*; *Chirillo et al., 2019*). Spines containing SER are larger than those without SER, and in adults 2 hr after induction of LTP the SER was elaborated into a spine apparatus in spines with enlarged synapses (*Chirillo et al., 2019*). Spines clustered around the enlarged spines and local shaft SER remained complex, whereas distant clusters had fewer spines than control dendrites and lost local shaft SER. These findings suggest that mature dendrites support a maximum amount of synaptic input and strengthening of some synapses uses resources that would otherwise be targeted to support spine outgrowth, even in adults.

At P15, CA1 dendrites have less than one-third mature synaptic density, which will nearly reach adult levels in another week (*Kirov et al., 2004*). These findings suggest that P15 may well be an age when synaptogenesis predominates over the growth of existing synapses, which may account for the spinogenesis response to LTP. At P15, SER was also restricted to a small number of spines, and like adults the few spines that had SER were larger than those without SER (*Chirillo et al., 2019*). However, at P15, most of the small, presumably newly formed spines did not contain SER.

Similar to adults, shaft SER was reduced in complexity and volume, but at P15 the redistribution was apparently targeted only to the plasma surface, rather than elaboration of the spine apparatus and growth of potentiated spines, as in adults (*Chirillo et al., 2019*). These findings suggest that synapse growth occurs where synapses had already been activated or previously potentiated, and few of those existed at P15 prior to the induction of LTP. Thus, resources were available for spine outgrowth to dominate. Future work is needed to learn when the shaft SER recovers, and when this recovery becomes necessary for additional synaptogenesis or synapse enlargement as the animals mature.

SER regulates intracellular calcium ion concentration (*Verkhratsky, 2005*). Regulation of postsynaptic calcium levels is necessary for the expression of synaptic plasticity (*Lynch et al., 1983*; *Malenka et al., 1988*), hence the presence of SER could be important for signaling cascades associated with LTP and stabilization of AMPA receptors at potentiated synapses (*Borgdorff and Choquet, 2002*). Consistent with this, spines with larger synapses tended to contain SER, and were maintained at stable density post-TBS. However, it might be of some concern that calcium regulation is disrupted by the reduction in SER volume in both adult and P15 hippocampal dendritic shafts by 2 hr after induction of LTP. The reduction in SER volume was by no means complete, and instead likely reflects the multiple roles of SER in membrane and protein trafficking in addition to the regulation of calcium. That a substantial amount of shaft SER remains well after the induction of LTP, supports the hypothesis that SER is a dynamically regulated resource at both ages.

### Role of satellite secretory system in synaptogenesis and subsequent plasticity

Dendrites support local processing and secretory trafficking of newly synthesized cargo independent of a Golgi apparatus (*Bowen et al., 2017*). Secretory cargo passes from the ER to ER-Golgi intermediate compartments (ERGICs) into recycling endosomes en route to the plasma membrane. While molecular understanding of this pathway has been achieved, the spatial organization of the responsible organelles has been nebulous. Recycling endosomes were seen about 25% of spines on cultured neurons that also contained synaptopodin, a marker for the ER-derived spine apparatus (*Bowen et al., 2017*). This finding suggested that recycling endosomes might receive newly synthesized cargo directly from a spine apparatus. However, at P15, only one spine apparatus was found in each of the control and TBS conditions, suggesting that recycling endosomes derive from alternate recycling organelles in the dendritic shaft. Previously, this satellite secretory system has only been studied in neurons under baseline conditions in culture. Here, we provide the first evidence that this specialized secretory system locally supports spine formation during plasticity.

Synaptogenesis at P15 does not precede the expression of LTP, as evidenced by a lack of added spines at 5 min following TBS (*Watson et al., 2016*). The magnitude of potentiation following the initial TBS is constant across time, so the added small spines at 2 hr after the induction of LTP are likely to be functionally silent. Hence, the newly added spines could be viewed as a form of heterosynaptic plasticity that readies the neurons for subsequent potentiation. In support of this hypothesis, a second bout of TBS delivered 90 min after the first TBS produces substantial additional potentiation at this age (*Cao and Harris, 2012*). Many of the added small spines contained endosomes at 2 hr after the initial induction of LTP. These endosomes might be interpreted as a heterosynaptic mechanism for long-term depression, namely internalizing receptors from pre-existing spines. However, since most of the endosomal structures occupied the added small spines and were of a constructive nature, they could instead be available to convert the new silent synapses to active synapses after a later bout of potentiation. Such a mechanism would support the establishment of functional circuits as the young animals learn and begin to form memories.

## Materials and methods

**Key resources table**

| Reagent type (species) or resource | Designation | Source or reference | Identifiers | Additional information |
| --- | --- | --- | --- | --- |

*Continued on next page*

*Continued*

| Reagent type (species) or resource | Designation | Source or reference | Identifiers | Additional information |
|---|---|---|---|---|
| Strain, strain background (Rattus norvegicus, male) | Long-Evans Rat | Charles River | Charles River strain# 006; RRID:RGD_2308852 | |
| Chemical compound, drug | Potassium ferrocyanide | Sigma-Aldrich | Cat# P3289 | |
| Chemical compound, drug | Osmium tetroxide | Electron Microscopy Sciences | Cat# 19190 | |
| Chemical compound, drug | Uranyl acetate | Electron Microscopy Sciences | Cat# 22400 | |
| Chemical compound, drug | LX-112 embedding kit | Ladd Research Industries | Cat# 21210 | |
| Chemical compound, drug | Lead nitrate | Ladd Research Industries | Cat# 23603 | |
| Chemical compound, drug | Pioloform F | Ted Pella | Cat# 19244 | |
| Software, algorithm | Igor Pro 4 | WaveMetrics | https://www.wavemetrics.net/ | |
| Software, algorithm | Reconstruct | *Fiala, 2005* | Executable and manual: http://synapseweb.clm.utexas.edu/software-0 | Source at: https://github.com/orgs/SynapseWeb/teams/reconstruct-developers |
| Software, algorithm | STATISTICA 13 Academic | Tibco | https://onthehub.com//statistica/ | |
| Other | Tissue slicer | Stoelting | Cat # 51425 | |
| Other | Vibratome | Leica Biosystems | VT1000S | |
| Other | Ultramicrotome | Leica Biosystems | UC6 | Used with a Diatome Ultra35 knife |
| Other | SynapTek Grids | Ted Pella | Cat# 4514 or 4516 | |
| Other | Diffraction grating replica | Electron Microscopy Sciences | Cat# 80051 | |
| Other | Transmission electron microscope | JEOL | JEM-1230 | |
| Other | Harris Lab wiki | Harris Lab | https://wikis.utexas.edu/display/khlab/ | This wiki site hosts experimental methods used for this paper and updates. |

## Animals

Hippocampal slices (400 µm) were rapidly prepared from P15 male Long-Evans rats (RRID:RGD_2308852, n > 100, including the initial test experiments and slices used in prior work for the 5 min and 30 min time points; *Watson et al., 2016*). For the 2 hr time point reported here, one slice each from two rats met the strict physiology and ultrastructural criteria for inclusion as outlined below. All procedures were approved by the University of Texas at Austin Institutional Animal Care and Use Committee and were followed in compliance with NIH requirements for humane animal care and use

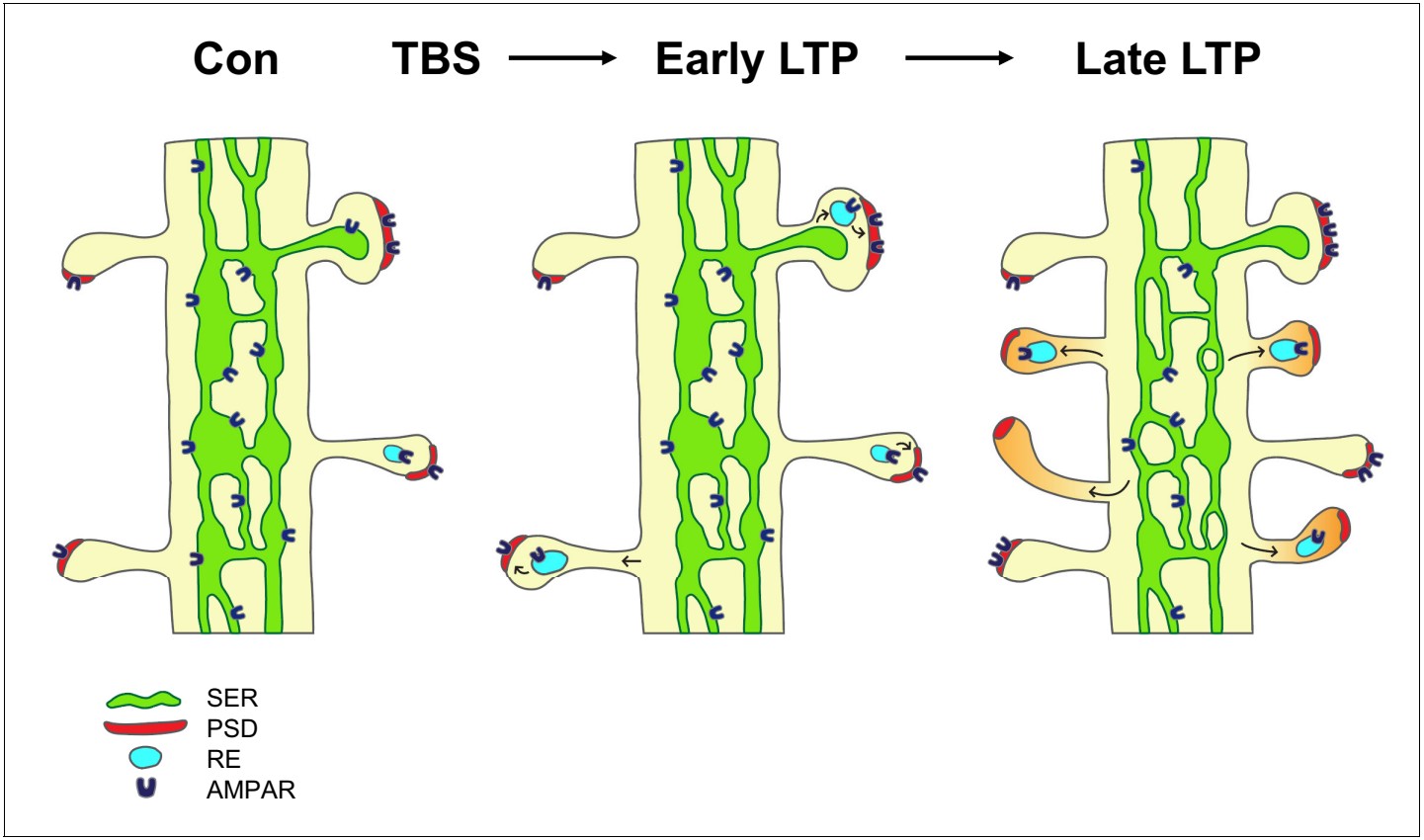

**Figure 6.** Model of the contribution of dendritic secretory compartments to LTP-induced synaptogenesis. Smooth endoplasmic reticulum (SER, green), postsynaptic density (PSD, red), small vesicle or recycling endosome (RE, turquoise), new silent spines (orange), control activation (Con), theta-burst stimulation (TBS), long-term potentiation (LTP), AMPA receptors (AMPAR).
DOI: https://doi.org/10.7554/eLife.46356.024

(Protocol number 06062801). All rats were of comparable features indicative of health at the time they were taken for experimentation.

## Preparation and recording from acute hippocampal slices

Rats were decapitated and the left hippocampus was removed and sliced into 400 µm thick slices from the middle third of the hippocampus at a 70° traverse to the long axis using a tissue chopper (Stoelting, Wood Dale, IL). Hippocampal slices were kept room temperature (~25°C) in artificial cerebrospinal fluid (ACSF) bubbled with 95% $O_2$/5% $CO_2$ (*Bourne et al., 2007*). ACSF consisted of 116.4 mM NaCl, 5.4 mM KCl, 3.2 mM $CaCl_2$, 1.6 mM $MgSO_4$, 26.2 $NaHCO_3$, 1.0 mM $NaH_2PO_4$, and 10 mM D-glucose at pH 7.4. Slices were immediately transferred to nets on top of wells containing ACSF at the interface of humidified $O_2$ (95%) and $CO_2$ (5%). Dissection and slice preparation took less than 5 min. The slices were kept at 32°C for approximately 3 hr *in vitro* prior to recording (*Fiala et al., 2003*). Two concentric bipolar stimulating electrodes (100 µm diameter, Fred Haer, Brunswick, ME) were positioned ~300–400 µm on either side of a single glass extracellular recording electrode in the middle of stratum radiatum for independent activation of subpopulations of synapses (*Sorra and Harris, 1998*; *Ostroff et al., 2002*; *Bourne and Harris, 2011*). The recording electrode was a glass micropipette filled with 120 µM NaCl. After initial recovery period, stable baseline recordings were obtained from both sites for a minimum of 40 min. Extracellular field potentials (fEPSPs) were obtained with custom designed stimulation data acquisition protocols using Igor software (WaveMetrics, Lake Oswego, OR). fEPSPs were estimated by linear regression over 400 µs along maximal initial slope (mV/ms) of test pulses of 100 µs constant, biphasic current. Stimulus

intensity was set to evoke 1/2 maximum fEPSP slope based on a stimulus/response curve for each experiment and was held constant for the duration of the experiment.

## TBS-LTP paradigm

Theta burst stimulation (TBS) was used to induce LTP. TBS was administered by one stimulating electrode as one episode of eight trains 30 s apart, each train consisting of 10 bursts at 5 Hz of 4 pulses at 100 Hz. The control stimulating electrode delivered one pulse every 2 min. Stimulations were alternated between the TBS-LTP and the control electrode once every two minutes with a 30 s interval between electrodes. In order to counterbalance across experiments, control and TBS-LTP electrode positions were interchanged between the CA3 and subicular side of the recording electrode (*Figure 1A*). Physiological responses were monitored for 120 min after the first train of TBS (*Figure 1B,C*) and then rapidly fixed, as described below.

## Fixation and processing for 3DEM

One slice from each animal was fixed and processed for electron microscopy 2 hr after induction of LTP. Only slices with good physiology were used, defined as a gradually inclining I/O curve in response to incremental increases in stimulus intensity for both stimulating electrodes, a stable baseline response at both stimulating electrodes unchanged at the control site post LTP-induction, and a significant increase in fEPSP slope that was immediately induced by TBS and was sustained for the duration of the experiment. Within a few seconds of the experiment's end, electrodes were removed and slices were immersed in fixative (6% glutaraldehyde and 2% paraformaldehyde in 100 mM cacodylate buffer with 2 mM $CaCl_2$ and 4 mM $MgSO_4$), microwaved at full power (700 W microwave oven) for 10 s to enhance penetration of fixative (*Jensen and Harris, 1989*), stored in the fixative overnight at room temperature, rinsed three times for 10 min in 100 mM cacodylate buffer, and embedded in 7% low melting temperature agarose. They were then trimmed, leaving only the CA1 region that contained the two stimulating electrodes. They were mounted in agarose and vibrasliced into 70 µm thick slices (VT1000S, Leica, Nusslock, Germany). Vibra-slices were kept in a 24-well tissue culture dish and examined under a dissecting microscope to locate the vibra-slices containing indentations from the stimulating electrodes.

The vibra-slices with the indentations due to the stimulating electrodes and two vibra-slices on either side of these indentations were collected and processed in 1% $OsO_4$ and 1.5% potassium ferrocyanide in 0.1M cacodylate buffer for 5–10 min, rinsed five times in buffer, immersed in 1% $OsO_4$ and microwaved (1 min on, 1 min off, 1 min on) twice with cooling to 20°C in between, and rinsed five times in buffer for two minutes and then twice in water. They were then dehydrated in ascending concentrations of ethanol (50%, 70%, 90%, and 100%) with 1–1.5% uranyl acetate and microwaved for 40 s at each concentration. Finally, slices were transferred through room temperature propylene oxide, embedded in LX-112 (Ladd Research, Williston, VT), and cured for 48 hr at 60°C in an oven (*Harris et al., 2006*).

Slices with high-quality preservation, defined as dendrites with evenly spaced microtubules, well-defined mitochondrial cristae, and well-defined PSDs that were not thickened or displaced from the membrane, were selected for analysis. The region of interest was selected from middle of the CA1 stratum radiatum and 120–150 µm beneath the air surface, then cut into 150–200 serial sections. The sections were mounted on Pioloform-coated slot grids (Synaptek, Ted Pella, Redding, CA). The sections were counterstained with saturated ethanolic uranyl acetate, then Reynolds lead citrate (*Reynolds, 1963*) for five minutes each, and then imaged with a JEOL JEM-1230 transmission electron microscope with a Gatan digital camera at 5000X magnification along with a diffraction grating replica for later calibration (0.463 µm cross line EMS, Hatfield, PA or Ted Pella). Imaging was conducted blind to condition.

## 3D reconstructions and measurements of dendrites

A random five-letter code was assigned to each series of images for the experimenter to be blind to the original experimental conditions during data collection. Reconstruct software (freely available at http://www.synapseweb.clm.utexas.edu; *Fiala, 2005*) was used to calibrate pixel size and section thickness, align sections, and trace dendrites, SER, endosomes, and PSD. The diffraction grating replica imaged with each series was used to calibrate pixel size. Cylindrical diameters method was used

to calculate section thickness (*Fiala and Harris, 2001*). Calculated section thicknesses ranged from 46 to 63 nm. Dendrites selected for analysis were chosen based on their orientation (cross-sectioned or radial oblique) and matched for diameter. Microtubule count was used as a measure of dendritic caliber (6–22 MTs) as this range under control condition showed no differences in spine density. All dendrites chosen for the analysis were completely reconstructed. The z-trace tool in Reconstruct was used to measure dendrite lengths across serial sections of each analyzed dendrite. Four dendrites were sampled from each condition (control or TBS-LTP) in each animal, resulting in a total of 16 dendritic segments from four EM series. Each analyzed dendritic segment traversed over 100 serial sections. In total, 173 µm of dendritic length was sampled.

## Identification and quantification of subcellular compartments

The process of tracing, reviewing, and curating dendrites, synapses, and subcellular objects was confirmed by three scientists (Kulik, Watson, and Harris) and conducted blind as to condition. On the rare occasions where there was disagreement, we met to arrive at a consensus based on the 3D structures; hence all objects were eventually provided a confirmed identification as outlined below.

Dendrites and PSDs were traced and dimensions were quantified as previously described (*Watson et al., 2016*). SER was identified on the basis of its characteristic morphology of tubules with dark staining membrane, occasional flattened cisternal distensions with a wavy membrane and clear lumen, and continuity across sections within each reconstructed dendrite. Once SER was completely traced, the remaining membrane-bound intracellular compartments were traced and their identity was assigned on the basis of morphology, as described in Results. Criteria used to differentiate endosomes included: 1) Continuity across sections: vesicles appear on single sections; tubules span multiple sections and then terminate; SER is continuous across sections throughout the entire dendrite; MVBs and tubules form a sorting complex when found on continuous sections; 2) Geometry: small and large vesicles are spherical, while amorphous vesicles are not; tubules have a uniform diameter across sections; SER has a highly variable profile across sections; MVBs have an unmistakable outer membrane surrounding multiple internal vesicles, and MVBs have tubules attached when part of a sorting complex; 3) Dimensions: small vesicles are 40–60 nm in diameter; large vesicles are 60–95 nm in diameter; 4) Electron density: amorphous vesicles and SER have a clear lumen; tubules and MVBs have a dark, grainy interior; lysosomes have a very dark, electron-dense interior.

Only spines that were entirely contained within the series were used for the analyses of subcellular compartments. In this way, we avoided possible undercounting of compartments that may have entered a portion of an incomplete spine outside the series. Spines were considered to contain a subcellular structure when it entered the head or neck of the spine, but not if it was only at the base of a spine. The frequency of occurrence was calculated as the total number of occurrences of objects divided by the length of dendrite in microns. The 3D visualization of dendrites and subcellular structures was achieved with Reconstruct. The 3D reconstructions from serial EMs allowed us to calculate volumes and surface areas of objects and to assess SER and endosome distribution within dendrites.

## Statistical analyses

The statistical package STATISTICA (version 13.3; TIBCO, Palo Alto, CA) was used for all analyses. There were two conditions represented in each animal: control (CON), and LTP at 120 min following TBS. In this study, eight control dendrites (four from each animal) and 8 LTP dendrites (four from each animal) were analyzed. One-way ANOVAs were run on all density (#/µm) data involving one measurement per dendrite, in which case n = number of dendrites. Hierarchical nested analysis of variance (hnANOVAs) were run when many measures were obtained from each dendrite (e.g. SER volume per spine, PSD area etc.). In this case, n = total spines, as each spine was considered separately. In hnANOVAs dendrite was nested in condition and experiment, and experiment nested in condition to account for inter-experiment variability. Results of the one-way ANOVAs and hnANOVAs are reported as ($F_{\text{(df condition, df observations)}}$=F value, P value) where df is degrees of freedom presented for condition and error. In hnANOVAs degrees of freedom are further decreased by one for each dendrite. Absolute p values are reported for each test. Statistical tests are reported in the figure legends. Data in bar graphs is plotted as mean ± SEM. Significant P values are indicated by asterisks above the bars. Significance was set at $p < 0.05$. The effect sizes for significant differences

are also presented in the figure legends as $\eta^2$ (which was determined as $SS_{condition}/SS_{(condition\ +\ error)}$, where SS = sum of squares determined in Statistica for each analysis).

We have provided the raw images, Reconstruct trace files, and analytical tables in the public domain at Texas Data Repository: DOI: https://doi.org/10.18738/T8/5TX9YA.

## Caveats

One might be concerned that these data arise from two animals. We note that these experiments are within-slice experiments, namely the control and LTP sites are from independent locations within the same slice from two different animals. Based on numerous preliminary experiments, we found that this approach greatly reduces variation due to slice preparation, *in vitro* conditions, and subsequent processing for electron microscopy when comparing the control and LTP outcomes. We also note that enhanced statistical power came from the large number of synapses and spines tested using the hierarchical nested ANOVA design with dendrite nested in condition by animal (*Figures 2E,F,H,I* and *3E*). In this way, degrees of freedom are adjusted for animal and dendrites, and outcomes are tested to ensure that no one dendrite or animal carried the findings. In addition, we had power to detect changes using multifactor ANOVAs for measurements that involved one measure per dendrite (#/μm listed on the y axes of *Figures 2B–D, G, 3C–D* and *5B–E*). Given the extremely time-consuming nature of the imaging and 3DEM analysis, additional animals and slices were not included.

## Source data files (Named Figures 1-5–source data 1 in each legend)

There is one source data file for each of *Figures 1–5* that contains Excel spreadsheets with the object locations in the Reconstruct trace files (provided in the public domain) for EMs. These files also contain the raw numbers that generated graphs in each part of each figure along with the summary of statistics.

## Acknowledgements

We thank Robert Smith and Elizabeth Perry for technical support in the ultramicrotomy; Heather Smith and Patrick Parker for their contributions in some of the dendrite analyses; and Patrick Parker for editorial comments. We thank Graeme W Davis for his support of YDK during the writing of this manuscript.

## Additional information

### Funding

| Funder | Grant reference number | Author |
| --- | --- | --- |
| National Institutes of Health | NS21184 | Kristen M Harris |
| National Institutes of Health | R01NS074644 | Kristen M Harris |
| National Institutes of Health | R01MH095980 | Kristen M Harris |
| National Institutes of Health | R01MH104319 | Kristen M Harris |
| National Science Foundation | NeuroNex 1707356 | Kristen M Harris |
| National Institutes of Health | F32 MH096459 | Deborah J Watson |

The funders had no role in study design, data collection and interpretation, or the decision to submit the work for publication.

### Author contributions

Yelena D Kulik, Conceptualization, Data curation, Formal analysis, Funding acquisition, Validation, Investigation, Visualization, Writing—original draft, Writing—review and editing; Deborah J Watson, Conceptualization, Data curation, Formal analysis, Supervision, Funding acquisition, Validation, Investigation, Visualization, Methodology, Writing—review and editing; Guan Cao, Conceptualization, Formal analysis, Validation, Investigation, Methodology, Writing—review and

editing; Masaaki Kuwajima, Data curation, Validation, Methodology, Writing—review and editing; Kristen M Harris, Conceptualization, Data curation, Formal analysis, Supervision, Funding acquisition, Validation, Investigation, Visualization, Methodology, Writing—original draft, Project administration, Writing—review and editing

### Author ORCIDs
Yelena D Kulik (iD) https://orcid.org/0000-0001-9543-532X
Guan Cao (iD) http://orcid.org/0000-0001-6211-5872
Masaaki Kuwajima (iD) https://orcid.org/0000-0002-1478-3726
Kristen M Harris (iD) https://orcid.org/0000-0002-1943-4744

### Ethics
Animal experimentation: All procedures were approved by the University of Texas at Austin Institutional Animal Care and Use Committee and were in compliance with NIH requirements for humane animal care and use. Protocol number (06062801). All rats were of comparable features indicative of health at the time they were taken for experimentation.

### Decision letter and Author response
Decision letter https://doi.org/10.7554/eLife.46356.029
Author response https://doi.org/10.7554/eLife.46356.030

## Additional files

### Supplementary files
• Transparent reporting form
DOI: https://doi.org/10.7554/eLife.46356.025

### Data availability
The relevant image series files and numerical data have been provided. In addition, the program Reconstruct is freely available from http://synapseweb.clm.utexas.edu/, and can be used to image and visualize the raw trace files. We have provided the raw images, Reconstruct trace files, and analytical tables in the public domain at Texas Data Repository: DOI: https://doi.org/10.18738/T8/5TX9YA.

The following dataset was generated:

| Author(s) | Year | Dataset title | Dataset URL | Database and Identifier |
|---|---|---|---|---|
| Kulik YD, Watson DJ, Cao G, Kuwajima M, Kristen M Harris | 2019 | Raw images, Reconstruct trace files, and analytical tables | https://doi.org/10.18738/T8/5TX9YA | Texas Data Repository, 10.18738/T8/5TX9YA |

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
