## [Decision Letter]

Thank you for submitting your article "Structural plasticity of dendritic secretory compartments during LTP-induced synaptogenesis" for consideration by *eLife*. Your article has been reviewed by Eve Marder as the Senior Editor, a Reviewing Editor, and three reviewers. The reviewers have opted to remain anonymous.

The reviewers have discussed the reviews with one another and the Reviewing Editor has drafted this decision to help you prepare a revised submission.

The reviewers agreed that the manuscript contains a set of very valuable data about structural changes in the setup of spines undergoing plasticity in young animals, with a particular focus on the smooth endoplasmatic reticulum and endocytic compartments.

The reviewers however raised two key concerns, which we request to be addressed in the revised manuscript:

- The reviewers are concerned about the underlying statistics that give rise to the conclusions drawn. In particular, the number of animals, slices, and reconstructed dendrites used are both unclear from the manuscript and suspected to be very small. Here, the reviewers request clarification and (if the low n is in fact confirmed) addition of further data to exclude inter-individual variability as a source of the observed effects.

- The methodological description is considered improvable by two reviewers who request better definition of concepts such as "new spine". For data annotations that may be considered subjective, the independent annotation by multiple experts may offer a way to provide classifications with confidence intervals.

Please see the detailed comments below for the context of these two key requests. All other comments can be treated as recommendations for the revision.

*Reviewer #1:*

This manuscript by Kulik et al. describes detailed ultra-structural analysis of dendritic spines after LTP using 3D reconstruction of serial section electron microscopy. They used relatively young animals, in which stage LTP induces spinogenesis. In particular, they focused on intracellular membrane structures including smooth endoplasmic reticulum (SER) and various endosomes.

They observed that spine density almost doubled 2 hours after LTP induction. Several differences in intracellular membrane structure in LTP area and control area are described. First, they observed less fraction of small spines after LTP contain SER, while large spines have reduced SER content. Next, they observed SER in dendritic shafts decreases their volume after LTP. Finally, spines containing various endosomes increased after LTP.

Overall, the manuscript contains potentially important data describing anatomical changes induced by LTP in young animals. However, some of their interpretation, for example their definition of "new spines" and their model of SER/endosome structure at the early time point, appears to be not well validated by the measurement.

More serious issue is that it is not clear how slice-to-slice and animal-to-animal variation can be taken into account. It appears that only two slices (not clear if they are from the same animal or different animals) are used in their analysis. The conclusion may not be generalizable to all animals.

Essential revisions:

1) Figure 1: The title "New small spines induced by TBS contained no SER while existing large spines had reduced SER content" sounds to be misleading. It is not described how they conclude that these are "new" spines. Also, I don't see any evidence suggesting "no SER" in "new spines" from the figure, as there are some SER in small spines.

2) Figure 5: "Increased endosomal activity" may be misleading, as they are not measuring the activity but the distribution of endosomes.

3) Statistics: It appears that the entire data is based on only two slices (Figure 1:. Is it from two animals?) Perhaps any statics would not work well with n=2. As LTP varies fairly a lot from slice to slice and from animal to animal, this raises a question of whether the conclusion can be generalizable.

4) Figure 6: It is not clear how they come up with the model of SER structure and endosome structure at "early LTP", as the measurement only at 2 hours.

*Reviewer #2:*

In this study, the authors conducted 3D EM reconstruction of CA1 dendrites after TBS LTP and concentrated on measuring organelles such as ER and endocytic compartments. Using this method, the authors make a few new observations:

1) New spines that are formed after LTP do not contain SER and existing spines lose some SER.

2) After LTP SER in dendritic shafts is reduced.

3) After LTP, increased endocytic compartments were observed in small spines.

Based on these observations the authors suggest that new spines observed after LTP are supported by recycling endosomes rather than SER. The apparent increase in endocytic structures after LTP is intriguing, although I have some concerns on how these structures are classified.

Essential revisions:

1) The identification of the specific endocytic structures in Figure 4 and Figure 5 relies solely on morphology, and based on a limited set of images it is unclear how reliable the distinction can be made between recycling endosomes and endosomes that may be heading to a lysosome/degradation pathway. The authors list a few papers as explanation of how these structures were classified, but the description of how the dendrites were annotated is vague. The authors need to provide a lot more detail on exactly how these structures were classified and how reliable is the distinction between similar-looking endocytic structures. Based on the current data presented, the conclusion that small spines are mostly supported by recycling endosomes is not strongly supported.

2) The 3D reconstruction of one time point after LTP makes it hard to infer dynamics from a static snapshot. While the reconstruction is extremely consuming, the authors need to discuss caveats and alternative interpretations of the data in the discussion section. For example, can the authors rule out heterosynaptic LTD or other homeostatic mechanisms that may rely on endocytosis of receptors?

*Reviewer #3:*

The paper by Kulik et al., examines the changes to the ultrastructure within dendrites of the P15 hippocampus after a stimulation protocol that produces long term potentiation. This piece of work is complementary to a number of other studies that have looked at similar changes in the adult and for this reason the paper it will be of some interest.

The study uses serial section EM to make detailed reconstructions of segments of dendrites that included the intracellular features such as smooth endoplasmic reticulum and endosomes. The analysis looked at dendritic spines, and the immediate piece of parent dendrite, as well as stretches of dendrite that bore no spines and compared their contents.

The stimulation protocol produced a significant increase in the number of dendritic spines, but not of spines that contained SER. The larger spines, however, considered as being the more permanent ones, had a reduction in their amount of SER, although still present. The authors conclude that these changes indicate that SER is not an organelle that is pivotal to the production of new spines induced by LTP. Measurements of the amount of SER in the parent dendrite show a decrease.

These changes led the authors to explore the intracellular ultrastructure in more detail and analyze the presence of different organelles. These they classed as either belonging to a degradative class such as lysosomes and multivesicular bodies, or constructive such as coated pits, vesicles, and endosomes. This analysis shows that the constructive elements were present only in the new (small) spines.

The study is well explained and illustrated. The quality of the electron microscopy is high and typical of this laboratory with considerable experience in this field of neuronal plasticity. The figures are well laid out and the descriptions are clear, as are the results and discussion parts.

The Materials and methods section fully describes the procedures used, including the details of the analysis. However, it would be useful if there is a clear explanation of how the different dendrites were sampled and grouped. It appears that two slices were used from two rats, and these produced 16 dendrites divided into two groups. It’s also stated that 'four 3DEM series were sampled'. Are 4 sets of serial images used? Or is it four different sets of sections from four different vibra-slices? It’s also not clear how many microns in dendritic length are sampled in total. Each dendrite from the 16 traverses over 100 serial sections. It would be useful to understand what sort of sampling has been done in this study.

For the analysis of the SER that is shown in Figure 3. The authors describe that there is less SER in the shaft after the LTP. There are less volume and less surface area as well as less cross-sectional surface area. As SER can have either a tubular or flattened appearance, I wonder whether these changes are due either to a simple alteration of the shape of the SER or a retraction of branches. Clearly, these are highly complex shapes, but it's not clear to me the significance of these results. Could a mere volume change account for the result rather than a removal of parts of the reticulum?

---

## [Author Response]

The reviewers agreed that the manuscript contains a set of very valuable data about structural changes in the setup of spines undergoing plasticity in young animals, with a particular focus on the smooth endoplasmatic reticulum and endocytic compartments.The reviewers however raised two key concerns, which we request to be addressed in the revised manuscript:Key concern 1: The reviewers are concerned about the underlying statistics that give rise to the conclusions drawn. In particular, the number of animals, slices, and reconstructed dendrites used are both unclear from the manuscript and suspected to be very small. Here, the reviewers request clarification and (if the low n is in fact confirmed) addition of further data to exclude inter-individual variability as a source of the observed effects.

Subsection “Caveats” has been added to the end of the Materials and methods section. We note that these experiments are within-slice experiments, namely the control and LTP sites are from independent locations within the same slice from two different animals. Based on numerous preliminary experiments, we found that this approach greatly reduces variation due to slice preparation, in vitro conditions, and subsequent processing for electron microscopy when comparing the control and LTP outcomes.

We also note that enhanced statistical power came from the large number of synapses and spines tested using the hierarchical nested ANOVA design with dendrite nested in condition by animal (Figure 2E,F,H,I, Figure 3E). In this way, degrees of freedom are adjusted for animal and dendrites, and outcomes are tested to ensure that no one dendrite or animal carried the findings. In addition, we had power to detect changes using multifactor ANOVAs for measurements that involved one measure per dendrite (#/µm listed on the y axes of Figure 2B-D, 2G, Figure 3C-D, Figure 5B-E). Given the extremely time-consuming nature of the imaging and 3DEM analysis, additional animals and slices were not included. (See subsection “Caveats”)

We have expanded the Materials and methods section to clarify the choice of animals, slices, reconstructed dendrites, spines, subcellular organelles, and statistics for these experiments. Regarding these choices the relevant text occurs at lines: Subsection “Animals”, subsection “Fixation and processing for 3DEM”, subsection “3D reconstructions and measurements of dendrites”, subsection “3D reconstructions and measurements of dendrites” and subsection “Statistical analyses”.

We have provided the F values, degrees of freedom, p values, and n’s in each Figure legend.

Where there were significant differences (p<0.05) we have added effect sizes (η^2^) and described how they were calculated in lines subsection “Caveats”.

We have added supplemental figures of complete 3D reconstructions of all the analyzed dendrites, arranged by condition and spine densities and illustrating the SER composition (Figure 2—figure supplement 1) or endosome composition (Figure 5—supplement 1 and Figure 5—supplement 2).

We have provided data source files for all of the figures.

We have prepared a site to release the raw images, reconstruct trace files, and analytical tables in the public domain at Texas Data Repository, DOI: https://doi.org/10.18738/T8/5TX9YA, which is not yet public, but will be upon acceptance of this paper.

Key concern 2: The methodological description is considered improvable by two reviewers who request better definition of concepts such as "new spine". For data annotations that may be considered subjective, the independent annotation by multiple experts may offer a way to provide classifications with confidence intervals.

Response regarding dynamic language such as “new spine”:

To address this concern, we have revised the wording in the Abstract, Results section and Discussion section to reflect our interpretations, and we added phrases like “LTP compared to control condition” to make clear that we have not actually watched the new spines form or other structures change, but made the interpretation.

For example, since there were three times as many small spines in the LTP as the control condition, we interpret this outcome to mean that a subpopulation of spines was added by 2 hours after the induction of LTP. We clarified these distinctions throughout the manuscript as follows:

Abstract only the last sentence uses “new” reflecting our model and interpretation.

Introduction restate the interpretation from the prior literature.

Results section provide an explicit explanation of how we arrived at the parsimonious explanation that some are new spines.

The Discussion section opens with a further explanation of how we arrived at the conclusion that there are new spines and puts the conclusions in the context of our model in Figure 6 and beyond.

Response regarding “subjective” annotations:

We added a new subsection “Identification and Quantification of subcellular compartments”.

We revised subsection “Identifying the dendritic trafficking network” to provide a more detailed description of the compartments including the revised Figure 4 legend.

New supplemental figures are provided that contain 3D reconstructions of all the dendrites to illustrate the SER in dendrites from the control and LTP conditions (Figure 2—figure supplement 1).

We added a diagram in Figure 4A that describes the definition of each endosome compartment, based on prior work using endocytosis of gold particles (from Cooney et al., 2002).

We added supplemental serial section images for each of the example endocytic compartments shown in Figure 4B-G and elaborated in Figure 4—figure supplement 1, Figure 4—figure supplement 2, Figure 4—figure supplement 3, Figure 4—figure supplement 4, Figure 4—figure supplement 5, Figure 4—figure supplement 6, which also include movies through serial sections when objects occupied more than 4 serial sections Figure 4—video 1, Figure 4—video 2, Figure 4—video 3.

As mentioned above, we are also prepared to release the raw images, reconstruct trace files, and analytical tables in the public domain.

Reviewer #1:

[…] Overall, the manuscript contains potentially important data describing anatomical changes induced by LTP in young animals. However, some of their interpretation, for example their definition of "new spines" and their model of SER/endosome structure at the early time point, appears to be not well validated by the measurement.More serious issue is that it is not clear how slice-to-slice and animal-to-animal variation can be taken into account. It appears that only two slices (not clear if they are from the same animal or different animals) are used in their analysis. The conclusion may not be generalizable to all animals.Essential revisions:1) Figure 1: The title "New small spines induced by TBS contained no SER while existing large spines had reduced SER content" sounds to be misleading. It is not described how they conclude that these are "new" spines. Also, I don't see any evidence suggesting "no SER" in "new spines" from the figure, as there are some SER in small spines.

The new title reads: Figure 2: The limited occupancy of spines by SER does not increase during spinogenesis in the LTP condition.

2) Figure 5: "Increased endosomal activity" may be misleading, as they are not measuring the activity but the distribution of endosomes.

Revised title reads: Figure 5: Increased occurrence of endosomes in small spines after LTP.

3) Statistics: It appears that the entire data is based on only two slices (Figure 1: Is it from two animals?) Perhaps any statics would not work well with n=2. As LTP varies fairly a lot from slice to slice and from animal to animal, this raises a question of whether the conclusion can be generalizable.

This comment is addressed above under Key concern #1.

4) Figure 6: It is not clear how they come up with the model of SER structure and endosome structure at "early LTP", as the measurement only at 2 hours.

This comment is addressed in the revised Discussion section, indicating that the earlier time points are deduced from the prior literature, more completely cited now and moved from the figure legend to the primary text of the Discussion section. Once the model is presented, then the subsequent sections have been revised to justify these interpretations – please see the revised Discussion section.

Reviewer #2:

In this study, the authors conducted 3D EM reconstruction of CA1 dendrites after TBS LTP and concentrated on measuring organelles such as ER and endocytic compartments. Using this method, the authors make a few new observations:1) New spines that are formed after LTP do not contain SER and existing spines lose some SER.2) After LTP SER in dendritic shafts is reduced.3) After LTP, increased endocytic compartments were observed in small spines.Based on these observations the authors suggest that new spines observed after LTP are supported by recycling endosomes rather than SER. The apparent increase in endocytic structures after LTP is intriguing, although I have some concerns on how these structures are classified.Essential revisions:1) The identification of the specific endocytic structures in figure 4 and Figure 5 relies solely on morphology, and based on a limited set of images it is unclear how reliable the distinction can be made between recycling endosomes and endosomes that may be heading to a lysosome/degradation pathway. The authors list a few papers as explanation of how these structures were classified, but the description of how the dendrites were annotated is vague. The authors need to provide a lot more detail on exactly how these structures were classified and how reliable is the distinction between similar-looking endocytic structures. Based on the current data presented, the conclusion that small spines are mostly supported by recycling endosomes is not strongly supported.2) The 3D reconstruction of one time point after LTP makes it hard to infer dynamics from a static snapshot. While the reconstruction is extremely consuming, the authors need to discuss caveats and alternative interpretations of the data in the discussion. For example, can the authors rule out heterosynaptic LTD or other homeostatic mechanisms that may rely on endocytosis of receptors?

Please see the complete response to Key concerns #1 and #2 above and revisions to the description of Figure 6 and the Discussion section that are also in response to these comments.

Specifically, regarding “heterosynaptic LTD and other homeostatic mechanisms”, we have added a few statements in the Discussion section, which describe a potential concern about calcium regulation if SER is diminished. We also discuss heterosynaptic LTD and explain why we opted for an interpretation that the constructive endosomes would have a positive impact on subsequent potentiation in the developing hippocampus, rather than reflect heterosynaptic LTD.

Reviewer #3:

[…] The study is well explained and illustrated. The quality of the electron microscopy is high and typical of this laboratory with considerable experience in this field of neuronal plasticity. The figures are well laid out and the descriptions are clear, as are the results and discussion parts.The Materials and methods section fully describes the procedures used, including the details of the analysis. However, it would be useful if there is a clear explanation of how the different dendrites were sampled and grouped. It appears that two slices were used from two rats, and these produced 16 dendrites divided into two groups. It’s also stated that 'four 3DEM series were sampled'. Are 4 sets of serial images used? Or is it four different sets of sections from four different vibra-slices? It’s also not clear how many microns in dendritic length are sampled in total. Each dendrite from the 16 traverses over 100 serial sections. It would be useful to understand what sort of sampling has been done in this study.

We have modified the methods as described under Key concern #1 above – specific to this inquiry please see subsection “3D reconstructions and measurements of dendrites” where we indicate that the total analyzed length was 173 µm. Also please note as indicated above that the n values are now given in each Figure legend. We also spell out that we compared the dendrites of comparable calibers, because spine density varies with dendrite caliber – see subsection “3D reconstructions and measurements of dendrites”.

For the analysis of the SER that is shown in Figure 3. The authors describe that there is less SER in the shaft after the LTP. There are less volume and less surface area as well as less cross-sectional surface area. As SER can have either a tubular or flattened appearance, I wonder whether these changes are due either to a simple alteration of the shape of the SER or a retraction of branches. Clearly, these are highly complex shapes, but it's not clear to me the significance of these results. Could a mere volume change account for the result rather than a removal of parts of the reticulum?

Yes, it is possible that a reduction in volume without a change in reticulum could have a similar impact. In fact, as shown in Figure 3, there was a lower SER volume and surface area in the LTP relative to the control condition. To address local complexity, we used the strategy developed in the Cui-Wang et al., 2012 paper, by summing the cross-sectional area of the SER profiles on a section by section basis. In this way, we were able to detect local variation and decrease in the combined volume and reticulum fractions and compare aspiny to spiny segments of the dendrite. We have clarified this description in subsection “Reduced complexity in shaft SER after LTP” regarding Figure 3 and added further comments regarding SER complexity in the Discussion section.